# Engineering Coinage Metal Nanoclusters for Electroluminescent Light-Emitting Diodes

**DOI:** 10.3390/nano12213837

**Published:** 2022-10-30

**Authors:** Tingting Li, Zhenyu Wang, Ying Zhang, Zhennan Wu

**Affiliations:** 1School of Materials Science and Engineering, Jilin Jianzhu University, Changchun 130018, China; 2State Key Laboratory of Integrated Optoelectronics, College of Electronic Science and Engineering, Jilin University, Changchun 130012, China; 3Department of Pediatric Respiratory, The First Hospital of Jilin University, Changchun 130012, China

**Keywords:** metal nanoclusters, light-emitting diodes, electroluminescent, emission layer

## Abstract

Coinage metal nanoclusters (MNCs) are a new type of ultra-small nanoparticles on the sub-nanometer (typically < three nm) scale intermediate between atoms and plasmonic nanoparticles. At the same time, the ultra-small size and discrete energy levels of MNCs enable them to exhibit molecular-like energy gaps, and the total structure involving the metal core and surface ligand together leads to their unique properties. As a novel environmentally friendly chromophore, MNCs are promising candidates for the construction of electroluminescent light-emitting diodes (LEDs). However, a systematic summary is urgently needed to correlate the properties of MNCs with their influences on electroluminescent LED applications, describe the synthetic strategies of highly luminescent MNCs for LEDs’ construction, and discuss the general influencing factors of MNC-based electroluminescent LEDs. In this review, we first discuss relevant photoemissions of MNCs that may have major influences on the performance of MNC-based electroluminescent LEDs, and then demonstrate the main synthetic strategies of highly luminescent MNCs. To this end, we illustrate the recent development of electroluminescent LEDs based on MNCs and present our perspectives on the opportunities and challenges, which may shed light on the design of MNC-based electroluminescent LEDs in the near future.

## 1. Introduction

As a new type of display and lighting device, electroluminescent light-emitting diodes have been widely integrated into people’s lives, owing to their high color purity, long lifetime, low power consumption, and other advantages [1,2,3,4,5]. Because of the advantages of having a narrower half-peak width, tunable emission in the entire visible spectrum, and higher color saturation, traditional colloidal semiconductor quantum dots (QDs) have been extensively studied in recent years [6,7,8]. However, QD materials generally contain heavy metal materials (Pb and Cd, etc.), which limits their large-scale commercial applications. Currently, there is an urgent need to develop new non-poisonous material for the light-emitting layer of electroluminescent LEDs, which is also of high application value.

Coinage metal (i.e., Au, Ag, and Cu) nanoclusters with atomically precise structures can be regarded as ultra-small metal nanoparticles (MNPs), aggregated from several to hundreds of metal atoms under the protection of monolayer organic ligands. They are a kind of metal atom congeries with a size typically below 3 nm, lying in the transition scale between the metal atoms and plasmonic metal nanoparticles [9,10,11,12,13]. Due to the ultra-small size of MNCs compared with MNPs, the original surface plasmon resonance effect disappears; in contrast, drastic changes occur in the energy levels of MNCs as a result of the particle size being close to the Fermi wavelength. That is, the electron levels of MNCs change from a quasi-continuous state to a discrete state due to the quantum size effect, which eventually exhibits molecular-like behaviors, such as the energy gaps from the discrete highest occupied molecular orbital (HOMO) to the lowest unoccupied molecular orbital (LUMO) and intense fluorescence [14]. Taking good photostability, easy preparation, low toxicity, etc. into consideration, the MNCs as the novel chromophore hold great potential in the applications of display and lighting. Most importantly, benefitting from the rapid development of synthetic chemistry, the total structure (i.e., the metal core, organic ligand shell, and their interface) of MNCs can be well regulated, which leads to their controllable and even customizable optical properties. This provides the basis for the fabrication of MNC-based electroluminescent LEDs [15,16,17,18,19,20]. However, one of the biggest disadvantages of MNCs is the relatively low photoluminescence quantum yields (PLQYs) compared to organic molecules, colloidal semiconductor QDs, carbon dots, and rare earth materials. To achieve the excellent performance of MNC-based electroluminescent LEDs, it is of great significance to prepare stable and highly luminescent MNCs, which has led to a tremendous number of studies on the emission origin and underlying structure-dedicated emission mechanism in luminescent MNCs over the past few decades [21,22,23,24,25]. Based on this background, we deploy this review to summarize the latest research progress on MNC-based electroluminescent LEDs. Specifically, an introduction to the influencing factors of the luminescence of MNCs, which is related to the LEDs’ performance, is first presented. On this basis, the synthesis methods of MNCs with high luminescence are reviewed. After that, the working principle and the current research progress of MNC-based electroluminescent LEDs are discussed. Finally, the existing problems and future development directions of MNC-based electroluminescent LEDs, based on our perspective, are clarified to provide guidance and direction for the research in this field.

As a new generation of lighting and display devices, LEDs have received extensive attention from scientific research and industrial circles in recent years [26,27,28]. According to different light-emitting principles, LEDs can be divided into two categories: photoluminescent LEDs and electroluminescent LEDs. Photoluminescent LEDs are passive luminescent devices, which use semiconductor chips as the light source and phosphor powder as the color conversion layer. They are mainly used for liquid crystal display panel backlights and daily lighting. Electroluminescent LEDs are autonomous luminescent devices, with the advantages of lighter, thinner, more power efficient, and flexible displays. According to the different materials in the light-emitting layer, electroluminescent LEDs can be divided into different types. Among them, organic light-emitting diodes (OLEDs) have been widely used. Meanwhile, owing to their wider color gamut and high color purity, quantum dots light-emitting diodes (QLEDs) have become a new research hotspot. Their photoelectric conversion performance in the laboratory can also be comparable to that of OLEDs, and they are in the commercial use and process development stage. However, due to the presence of toxic heavy metals, these LEDs may cause environmental pollution. It is of practical significance to develop a new type of stable and efficient electroluminescent LEDs based on non-toxic luminescent materials. Fluorescent MNCs, whose size is just between small organic molecules and QDs, have excellent molecular-like fluorescent properties, are environmentally friendly, and have great potential to be used as light-emitting layer materials [29,30,31,32,33,34,35,36]. Through developments in recent years, MNCs have shown the advantages of environmental protection and low cost when they have been applied to electroluminescent LEDs. However, compared with photoluminescent LEDs, electroluminescent LEDs are much more complex in structure and process, making their technical thresholds relatively high [37,38,39,40,41,42,43,44,45], especially since the light-emitting layer is generally only on a nanoscale, resulting in many AIE-enhanced fluorescent MNCs being too large to be used. The development of electroluminescent LEDs based on fluorescent MNCs is relatively lagging behind, and there are only a few related work reports. The latest research progress of MNCs in the field of electroluminescent LEDs will be introduced in the following paper. All abbreviations used in the text are shown in Table 1.

## 2. Correlating the Photoemission of MNCs to Their LED Applications

As shown in Figure 1a, MNCs are composed of metal core and organic ligand shell, and the metal core and organic shell greatly determine their physicochemical properties, which may further affect the design and performance of MNC-based electroluminescent LEDs. Therefore, understanding the physicochemical properties of MNCs is beneficial to designing high-performance LEDs based on MNCs. The high PLQYs of MNCs as the emissive layer is crucial to achieving an excellent performance for devices of superior electroluminescent LEDs based on MNCs. After years of continuous research, several effective strategies have been developed to obtain highly luminescent MNCs, including ligand engineering, surface motif engineering, heterometal doping, and structural modulation [46,47,48]. In this section, methods to synthesize highly luminescent MNCs will be discussed, focusing on the luminescence enhancement mechanism.

### 2.1. Metal Core-Related Photoemission

As the metal part of core-shell MNCs, the physicochemical properties of the metal core play an important role in the luminescence of MNCs as well as the performance of MNC-based electroluminescent LEDs.

#### 2.1.1. Effect of the Size of Metal Core

A very simple theoretical model to describe the photoluminescence of MNCs is the Jellium model, which is used to explain the effect of size-dependent quantum confinement effects on the photophysical properties of MNCs. According to the description of this model, the valence electrons of NCs are confined in orbitals with the same symmetry as the atoms, and are filled according to similar rules [49]. As the size of MNCs continuously increases to the range of MNPs, the optical energy gap (E_g_) of MNCs continuously decreases until the discrete molecule-like energy levels become quasi-continuous in the MNPs’ energy level state. However, the corresponding relationship between E_g_ and size is not a smooth trend of decay. As shown in Figure 2a, there is no linear proportional relationship between the E_g_ of MNCs and the number of metal atoms in the metal core. On the contrary, the relationship between E_g_ and n^−1/3^ exhibits a good linear scale and can be described by an empirical formula, E_g_ = an^−1/3^−b, where a and b are the fitting parameters [50]. Spherical MNCs can be predicted according to the formula, and the transition energies of MNCs are proportional to their radii. The model also predicts that as N increases, the energy level spacing decreases until it becomes smaller than K_b_T, at which point the plasmonic collective motion of electrons is allowed. This is a very simple theoretical model that does not take the exact structure of MNCs into account. It works well for general spherical MNCs. However, in addition to the size of the metal core, the composition and structure of the metal core can also significantly affect the luminescent properties of MNCs.

#### 2.1.2. Effect of the Oxidation State of Metal Core

The metal components of MNCs consist of two parts, including the zero-valent metal core and the univalent metal component located in the surface motifs. Under the combined action of these two parts, the overall valence state of the metal component of the MNCs is between 0 and 1, and the overall valence state of the MNCs changes simultaneously with the relative ratio of the metal core to the surface motifs. Usually, the relative increase in the monovalent metal component in the metal core will lead to the correction of the valence state of the metal components in the whole MNCs, and the ligand-to-metal charge transfer (LMCT) effect will be promoted, which is beneficial to enhancing the luminescence of the MNCs. In addition, the increase in the oxidation state of the metal cores is beneficial for improving the oxidation resistance of the MNCs. In 2012, Wu et al. reported that oxidation of Au_25_(SG)_18_ by H_2_O_2_ resulted in the significantly enhanced fluorescence of Au_25_ NCs [51]. After the addition of H_2_O_2_, the fluorescence intensity of the Au_25_ NCs was increased by 2.3-fold relative to the initial state, as shown in Figure 2b. The enhancement in the luminescence properties may be due to the alteration of the relative ratio of Au^0^ states in the metal core and Au^1^ in the surface motifs of the Au_25_ NCs.

#### 2.1.3. Effect of the Composition and Structure of Metal Core

The metal core of MNCs is usually composed of coinage metals and their alloys. The composition of MNCs also greatly affects their luminescence, and some alloy MNCs have recently been found to exhibit stronger luminescence than mono-metal NCs of the same size and structure. Doping metals with relatively low electronegativity into the metal cores of MNCs composed of high-electronegativity metals reduces the PLQYs; otherwise, the PLQYs can be enhanced. In 2016, Kang et al. reported changing the shape of spherical MNCs by employing a metal exchange method [52]. As shown in Figure 3a, the interaction of Pt_1_Ag_24_ with Au complexes coordinated by different ligands resulted in spherical and rod-shaped trimetallic NCs, respectively. Additionally, the PLQYs of the rod-like trimetallic NCs increased sharply from 0.1% before the metal exchange to 14.7%, which also proved that both the composition and shape of the metal core in MNCs have a great influence on the PLQY of the final product.

The incorporation of heterogeneous metal atoms into MNCs is a commonly used method to tune the luminescence properties of MNCs. The fluorescence QYs of MNCs can be improved by incorporating metal atoms with higher electronegativity. In general, alloyed MNCs exhibit stronger luminescence than their monometallic analogs due to the enhanced LMCT or ligand to metal–metal charge transfer (LMMCT) effects and changes in the electron/geometric structure of MNCs. The formation of alloy MNCs has become an important means to enhance the luminescence brightness of MNCs, and the fluorescence properties of MNCs can be significantly changed by adopting various approaches to alter the structure, position, and shape of the metal core. Heterometallic doping has been widely reported as a way to improve the luminescence properties of MNCs. One of the most common combinations is the incorporation of Au atoms into Ag NCs. Since the electronegativity of Au (2.54) is greater than that of Ag (1.93), the luminescence brightness of Ag NCs has been greatly enhanced. The study by Bakr et al. showed that doping a single Au atom at the center of Ag_25_ NCs can dramatically increase the PLQYs by a factor of ~25 [53]. Moreover, other combinations of relatively large electronegativity heteroatoms incorporated into MNCs have also been reported. By systematically studying single Pt/Pd/Au-doped Ag_25_ NCs, Wu et al. found that Au/Pt doping could improve the PLQYs of the clusters [54]. The experiments of Zhu et al. showed that doping Pt at the center of Ag_29_ NCs can effectively improve the optical QYs of the clusters [55]. In 2017, Du et al. reported that when Ag_50_(Dppm)_6_(SR)_30_ was doped with Au atoms, both the thermal stability and luminescence intensity of the obtained Au_x_Ag_50−x_(Dppm)_6_(SR)_30_ NCs were significantly improved [56], as shown in Figure 3b. It also demonstrates that heterometallic doping can significantly alter the optical properties of metal nanoclusters.

### 2.2. Ligand Shell-Related Photoemission

The ligand shell of MNCs is also important for their practical applications because it directly determines the interface behavior of MNCs during the application, and the surface ligand shells of MNCs can be designed to obtain the desired functions, which can significantly improve the interface behavior performance of MNCs in LED applications.

#### 2.2.1. Effect of the Structure and Steric Hindrance of the Ligand Shell

The structure and steric hindrance of MNC surface ligands have considerable influence on the performance of MNC-based LEDs, and the structure of MNC ligands can be flexible or rigid. If the surface ligands are rigid, the non-radiative energy loss caused by the vibration, stretching, and rotation of the ligands can be effectively suppressed. It is beneficial to the luminescence of metal clusters. At the same time, if the surface ligands have greater steric hindrance, it means that there are more uncoordinated metal atomic sites on the MNCs’ surface [57,58]. In 2017, Kang et al. reported the modulation of the fluorescence intensity of Au_2_Cu_6_ NCs by employing phosphine ligands with different electron-donating or electron-withdrawing substituents [59]. As shown in Figure 4a,b, when protected by electron-rich ligands, the LMCT is enhanced, and the luminescence of Au_2_Cu_6_ NCs is significantly enhanced, while the PLQY increases from the initial 12.2% to 17.7%. Meanwhile, when protected by electron-deficient ligands, the luminescence intensity of Au_2_Cu_6_ NCs is decreased, and the PLQY is reduced compared to the initial state.

Among several strategies to regulate the luminescence of MNCs, the most commonly used method is to modulate the ligands on the surface of the nanoclusters. In 2010, Jin reported that the LMCT process could enhance the luminescence of MNCs [60]. Both the surface ligands with electron-donating properties and the electron-rich atoms in the ligands promote the contribution of electrons to the metal core corresponding to the LMCT process. In 2016, Gan et al. synthesized three fluorescent Au_24_(SR)_20_ NCs with similar structures, all with metal cores protected by surface motif Au_4_(SR)_5_ and forming interlocking structures [61]. As shown in Figure 3e and f, the vibration of the surface ligand is suppressed, thus leading to a reduced energy loss from the nonradiative transition, while the surface ligands with electron-donating properties enhance the LMCT process, both of which jointly contribute to the fluorescence enhancement. This also demonstrates that the surface ligands significantly affect the luminescence properties of the NCs.

Based on the above, when designing high-performance light-emitting diodes based on MNCs, the influence of the ligand shell of MNCs in terms of structure and steric hindrance must be considered.

#### 2.2.2. Effect of the Tailorability of the Ligand Shell

Ligand shells of metal clusters with rich but customizable surface chemistry offer opportunities to improve the performance of LEDs, and researchers often tune the flexibility of MNCs’ ligand shells by grafting rigid molecules. By introducing higher electronegativity atoms or groups into surface motifs, the quantum yield of MNCs will increase, which is attributed to the enhanced LMCT effect. As shown in Figure 4c,d, Kang et al. found that by adding triphenylphosphine to Ag_29_(BDT)_12_(TPP)_4_ NCs, the dissociation process of surface ligands was restricted, while the PLQY of the NCs increased from 0.9% of the initial state to 11.7% [62]. The authors found that the photoluminescence intensity of Ag_29_(BDT)_12_(TPP)_4_ also showed a 25-fold enhancement relative to the initial state when the ambient temperature decreased, which was ascribed to the fact that the dynamic dissociation process of surface ligands was suppressed at a low temperature. This work also demonstrates that by tuning the surface ligands, the luminescence of nanoclusters can be significantly altered.

In recent years, the aggregation-induced emission (AIE) theory reported by Tang et al. was introduced into the field of MNCs to guide the synthesis of highly luminescent MNCs through surface engineering, leading to an explosion of progress [63,64]. Based on the AIE principle, the surface motifs of MNCs can enhance intracluster and intercluster metalophilic interactions with the help of solvents or ions, and reduce non-radiative energy loss by constraining ligand vibrations, significantly promoting the luminescence of MNCs.

Recently, the experiment by Liu et al. showed that as the content of the surface motif Au+-GSH in the MNCs changed, the luminescence peak position also achieved a significant change from 600 nm to 810 nm, as shown in Figure 5a [65]. Due to the electron-rich nature, the charge transfer between the metal and the GSH ligand is stronger when the content of the surface motif of the MNCs is higher, corresponding to the high energy emission of Au NCs at 600 nm; in contrast, the charge transfer between the metal and the ligand is weak when the content of the surface motif of Au NCs is lower, resulting in a low energy emission at 810 nm. Recently, Wu et al. showed that the length of the surface M(I)-SR motif can be regulated to alter the luminescence wavelength of the MNCs [66]. In particular, reducing the length of the Au+-SR motifs on the Au NCs’ surface can not only adjust the emission wavelength from the visible light to the NIR-II range but also transform the origin of Au NCs’ luminescence from the AIE characteristic phosphorescence to the Au(0) core. Furthermore, the authors emphasized that this effect would be more pronounced when Au NCs were aggregated to a greater extent in a solution.

### 2.3. Structure Modulation of MNCs

The luminescence properties of MNCs can be altered by adjusting the packing of metal core atoms. As shown in Figure 5b, Zhuang et al. reported two Au_42_(TBBT)_26_ NCs (TBBT = 4-tert-butyl sulfide), in which Au_42_ NCs with a face-center-cubic (fcc) structure exhibited higher luminescence brightness than Au_42_ NCs with a non-fcc structure [67]. Meanwhile, the luminescence of three other pairs of MNCs of similar sizes but different metal core packing modes was also compared, and these results all coincided with the experimental results above, indicating that the fcc packing metal core of MNCs may be more favorable for luminescence compared to the non-fcc structure. Recently, Zhou et al. reported that Au_38_ NCs with body-centered cubic (bcc) core packing showed ultra-long excited state lifetimes that were three orders of magnitude higher than those of hexagonal close-packed (hcp) Au_30_ NCs [68]. The two MNCs with different core packing methods comprise the same Au_4_ tetrahedral unit, and the origin of the huge difference in the excitation lifetimes between MNCs with different packing modes lies in the apparent change in the wave function of the Au_4_ tetrahedral unit. The two representative examples mentioned above clearly demonstrate that structural modulation can effectively modulate the luminescence of MNCs. However, progress in this area is at an early stage, and further research work is required to systematically explore the fine-structure modulation strategies of MNCs and fundamentally understand their structure–luminescence correlations.

To sum up, the above several pathways to synthesize highly luminescent MNCs have been extensively explored by researchers, and their effectiveness has been demonstrated. However, there have been few attempts to combine two or more synthetic routes to enhance the luminescence of MNCs. We believe that through the combination of the above strategies, MNCs with excellent luminescence properties suitable for the light-emitting layer of MNC-based electroluminescence LEDs will be reported in the future.

## 3. Electroluminescent Light-Emitting Diodes with Metal Nanoclusters

The device structure of MNCs based on electroluminescent LEDs is similar to that of traditional OLEDs, which is a typical carrier injection electroluminescent device with a “sandwich” structure, and MNCs’ thin film is used as the light-emitting layer [69,70,71,72]. As shown in Figure 6, electroluminescent LEDs are generally composed of a hole transport layer (HTL), a light-emitting layer, an electron transport layer (ETL), and electrodes, and corresponding interface modifications can be performed between the layers according to special requirements. Electroluminescent LEDs based on MNCs are generally divided into two structures: formal and trans structures. The light-emitting mechanism of both structures is the same and can be divided into three processes. First, driven by an external electric field, electrons and holes are injected from the cathode and anode to the LUMO energy level of ETL and the HOMO energy level of HTL, overcoming the interfacial barrier, respectively. Secondly, electrons and holes are transported into the MNC-based light-emitting layer driven by an electric field to recombine and form excitons. Finally, the excitons transition from the excited state back to the ground state, releasing energy in the form of light or heat [73,74,75]. The electroluminescence process should minimize the energy loss caused by the nonradiative transition process, thereby improving the luminescence efficiency. Due to the different energy levels and carrier transport capabilities of different MNCs, when they are used as light-emitting layers of LEDs, the performance of LEDs can be optimized by selecting transport layer materials with matching energy levels and adjusting the thickness of each layer.

### 3.1. Monometallic Nanoclusters

In 2014, Bjoern et al. reported the first monometallic NC-based electroluminescent LEDs [76]. Due to the poor film-forming properties of the MNCs in the aqueous phase, researchers used cetyltrimethylammonium bromide (CTAB) to transfer the MNCs from the aqueous phase to the toluene phase, and obtained glutathione-protected Ag and Au NCs with emission wavelengths at 697 nm and 750 nm, respectively. The device structure is shown in Figure 7a, and the obtained Ag and Au NCs are used as light-emitting layer materials to prepare electroluminescent LEDs. The external quantum efficiency (EQE) of the device reached the maximal value of 0.013% at the current density of 3.7 mA cm^−2^. As the voltage increases, the contribution from the hole injection layer (HIL) in the electroluminescence spectrum of the LED increases significantly, indicating that when the voltage reaches a certain value, the exciton recombination also occurs in the HIL in addition to the Au NCs’ light-emitting layer, and it is detrimental to the performance of the LEDs.

In 2015, Koh et al. further improved the performance of MNC-based LEDs from two aspects on the basis of previous work [77]. On the one hand, they used Au NCs with higher PLQYs in the selection of light-emitting layer materials. On the other hand, as shown in Figure 7b, by introducing a layer of ethoxylated polyethyleneimine (PEIE) as ETL between the ZnO electron injection layer (EIL) and the Au NCs’ light-emitting layer, the energy level of the device is optimized and the injection of electrons and holes are more balanced. Through the joint efforts of these two aspects, the maximal brightness of the device exceeds 40 cd/m^2^, while the maximum EQE reaches 0.12%, and the parasitic emission originating from HIL with increasing voltage as previously reported is eliminated. In addition, to be used as light-emitting layers alone, MNCs can also be mixed with other light-emitting materials as light-emitting layers. In 2018, Chao et al. mixed trioctylphosphine-protected yellow Au NCs with a blue-emitting organic host material as a light-emitting layer (the device structure is shown in Figure 7c), and successfully prepared a white light device with a current efficiency as high as 0.13 cd/A, color coordinates at (0.27, 0.33), and the maximum brightness reaching 100 cd/m^2^ [78]. The optical and electrical performances of the electroluminescent LEDs based on MNCs described in the text are summarized in Table 2.

These previous works confirmed that fluorescent MNCs can be used as the light-emitting layers of electroluminescent LEDs; however, the luminous efficiency of current devices is still very low, far from reaching the application level. The performance level of MNCs and the lack of device structure optimization have resulted in the current research situation, and there is still a large research space in this field.

### 3.2. Heterometallic Nanoclusters

Due to the strong spin–orbit coupling caused by multiple heavy metal atoms, the crossover from singlet to triplet states between systems is efficient, and MNCs exhibit some promising properties as potential emission layer materials for phosphorescent electroluminescent LEDs [91,92]. Furthermore, intramolecular metal–metal interactions comparable to hydrogen bond energies not only enhance the molecular rigidity and thus the phosphorescence efficiency of the MNCs, but also enhance the structural stability against thermal and radiative interference. Since MNCs are mostly ionic, the solubility and charge mobility of MNCs can be significantly improved by the judicious selection of soluble counterions with excellent carrier transport capabilities.

Phosphorescent electroluminescent LEDs based on Au(III), Ag(I), and Cu(I) NCs have been developed due to their outstanding applications and industrialization prospects in solid-state lighting and full-color fluorescent flat panel displays. In 2015, Xu et al. prepared cationic Au_4_Ag_2_ heterohexanuclear NCs with bis(2-diphenylphosphinoethyl) phenylphosphine as the ligand [79]. The Au_4_Ag_2_ NCs exhibit high-intensity phosphorescent emission in the solid state and thin films. Different from the general LED light-emitting layer with only light-emitting materials, the researchers mixed Au_4_Ag_2_ NCs, hole transport materials, and electron transport materials as the light-emitting layer, which is beneficial to solve the problem of the poor carrier transport ability of MNCs. As shown in Figure 8a, a layer of copper thiocyanate (CuSCN) was introduced as the HTL in the device, which optimizes the energy level of the device and reduces the potential barrier between energy levels. Under the joint optimization of these two parts, the maximal value of the EQE based on Au_4_Ag_2_ is 7.0%, and the maximal value of the current efficiency reaches 24.1 cd A^−1^.

In 2016, Xu et al. used rigid tris(diphenylphosphine)methane (CH(PPh_2_)_3_) as a surface ligand to obtain a quantum yield of 53% in a solution of CH_2_Cl_2_ and 78% in a powder state [80]. They produced triangular prism-shaped cationic Ag_6_Cu isoheptylnuclear alkynyl NCs in which the central copper(I) atom was wrapped by six silver(I) atoms through a silver–copper bond. Since these strongly phosphorescent Ag_6_Cu clusters not only have excellent photochemical and thermal stability but also are soluble in organic solvents, they are applied in solution-processed electroluminescent LEDs, in the light-emitting layer of the device, with hole transport materials tris(4-(9H-carbazol-9yl)phenyl)amine (TCTA) and using the electron transport material 1,3-bis(5(4-(tert-butyl)phenyl)-1,3,4-oxadiazol-2-yl)benzene (OXD-7) as a mixed host into which Ag_6_Cu clusters are doped. The device structure is shown in Figure 8b. The device based on the four-layer structure can achieve high-efficiency electroluminescence at a low voltage (4.5 V) with a current efficiency of 42.5 cd A^−1^ and an EQE of 13.9%.

Many planar mononuclear Pt(II) NCs do not emit light or are weakly emissive at room temperature, but their corresponding Pt(II)–M(I) (M=Ag, Au) heteronuclear NCs are more rigid due to the greatly enhanced molecular rigidity when forming aggregated structures. Accordingly, they exhibit highly phosphorescent properties. In 2018, Natarajan et al. synthesized a series of highly phosphorescent PtM_3_ (M=Au, Ag) aromatic acetyl NCs [81]. Due to the greater rigidity of the tetraphosphorus ligand, the intermolecular aggregation is greatly reduced, and the efficiency decay in the device is effectively suppressed. The significant difference between the PtAg_3_ and PtAu_3_ NCs’ structures stems from the position of the Pt atoms, and the difference in bonding between gold(I) and silver(I); the acetylide leads to the formation of a completely different PtM_3_ NCs’ structure. The phosphorescence emission of PtAu_3_ NCs is much stronger than that of PtAg_3_ complexes with a flow structure. PtAu_3_ clusters show strong phosphorescence with quantum yields over 90% in thin films. The device structure is shown in Figure 8c, and the EQE of electroluminescent LEDs based on PtAu_3_ NCs exceeds 18%. The optimized device achieves a maximum brightness of over 1000 cd m^−2^ and exhibits minimal efficiency degradation (less than 1%) at peak brightness. In 2019, Natarajan et al. utilized a series of structurally diverse aromatic acetylate ligands to tune the fluorescence emission properties of tetraphosphine-supported PtAu_3_ NCs [82]. The full width at half maximum (FWHM) of the phosphorescent emission is significantly affected by the dihedral angle between the arylacylate plane and the Pt(II) coordination square plane, in which the better the coplanarity, the narrower the FWHM. The structural differences in the ligands change the spatial arrangement of the PtAu_3_ NCs, which can tune the bandwidth of the emission spectrum. The PtAu_3_ NCs also showed bright phosphorescence with PLQYs as high as 80.5–90.1% in the films. Efficient electroluminescent LEDs were fabricated by doping different PtAu_3_ NCs into mCP and OXD-7 mixed host materials as emission layers, as shown in Figure 8d. Devices with a peak current efficiency (CE) of 38.7 cd A^−1^, EQE of 10.3%, FWHM of 42 nm, and peak CE of 62.2 cd A^−1^, EQE of 16.6%, and FWHM of 59 nm were achieved, respectively.

In 2020, Huang et al. used a rigid diformylcarbazole ligand, 3,6-di-tert-butyl-1,8-diformyl-9H-carbazole (H_3_decz), for highly soluble phosphorescent Ag-Au NCs that were rationally designed to prepare three Ag-Au NCs with special chain-ring structures driven by metallophilic interactions [83]. The self-assembly reaction of H_3_decz, Au^+^, and Ag^+^ produces anionic Ag_4_Au_6_ NCs-1. Among them, H_2_decz has an excellent hole-transporting ability and has a free ethynyl group. When the four free ethynyl groups in phosphorescent Ag_4_Au_6_ NCs-1 are further partially bound by four (PPh_3_)Au^+^ and four (PPh_3_)Ag^+^ units, the nonradiative ethynyl vibrations are eliminated and the formed Ag_8_Au_10_ NCs-2 exhibits stronger phosphorescence. After using diphosphate Ph_2_P(CH_2_)4PPh_2_ instead of PPh_3_, the phosphorescence efficiency in Ag_8_Au_10_ NCs-3 was further improved, and the phosphorescence efficiency in the doped film was as high as 96%. The device structure is shown in Figure 8e, and the CE and EQE of Ag_8_Au_10_ cluster-based electroluminescent LEDs are 47.2 cd A^−1^ and 15.7% (Ag_8_Au_10_ NCs-2) and 50.5 cd A^−1^ and 14.9% (Ag_8_Au_10_ NCs-3), respectively.

In 2021, Jiao et al. utilized Cu^+^, Ag^+^, carbazole ethynyl ligand 1-(3,6-di-tert-butylcarbazol-9-yl)-4ethynylbenzo and pyridyl-diphosphine ligand 2,6-bis(diphenylphosphino)pyridine in a self-assembly reaction that yielded a highly emissive Ag_3_Cu_5_ NCs [84]. The bisphosphine-supported Ag_3_Cu_5_ NCs are highly stable through a large number of Cu-Ag and Cu-Cu intermetallic interactions. Due to the effective shielding of Ag_3_Cu_5_ NCs’ centers by bulky ligands, the nonradiative relaxation of triplet excited states is minimized, providing excellent photoluminescence emission at ambient conditions with phosphorescence efficiencies of 18% in a solution and 75% in thin films, respectively. The electroluminescent LEDs based on Ag_3_Cu_5_ NCs’ films exhibit highly efficient yellow electroluminescence with a peak CE of 38.1 cd A^−1^ and EQE of 14.7%. The device structure is shown in Figure 8f.

### 3.3. Copper Iodide Nanoclusters

In recent years, luminescent copper iodide NCs have attracted extensive attention due to their high elemental crustal abundance, structural diversity, ease of preparation, and high PLQYs [93,94,95]. More importantly, the strong structural rigidity of the copper NCs can effectively compensate for the John-Teller twist-induced quenching of excited states and thermal/photodissociation of Cu^+^ ions. In this sense, copper iodide NCs have great potential as a novel high-efficiency and low-cost electroluminescent material. Copper(I) iodide NC luminescent materials have attracted widespread attention due to their low cost and high brightness, especially mononuclear and dinuclear copper iodide NCs. In recent years, breakthroughs have been made in the application of copper iodide NC-based electroluminescent LEDs, but there are still problems, such as energy loss and poor photostability caused by the relaxation of the excited state. Compared with mononuclear and dinuclear copper complexes, Cu_4_I_4_ NCs have extremely rigid structures and excellent photothermal stability, and have been used as outer coatings of photoluminescent LEDs to achieve excellent full-color and white light emission [96,97]. However, the processing properties of Cu_4_I_4_ NCs are insufficient, especially because the electrical properties are often weak, which makes it difficult to directly use them as the light-emitting layer material in electroluminescent LEDs.

In 2017, Hui Xu et al. prepared the first electroluminescent Cu_4_I_4_ NCs (DBFDP)_2_Cu_4_I_4_ through ligand engineering design, which solved the problems of poor processability and weak electroactivity of Cu_4_I_4_ NCs [85]. In the bidentate phosphine ligand 2,9-bis(diphenylphosphine)-dibenzofuran (DBFDP), the distance between the P-P bonds is 5.8 Å, which can just form a stable coordination junction with Cu_4_I_4_ NCs. The host properties of DBFDP significantly enhance the carrier injection and transport capability and improve the solubility. Additionally, they enhance the double emission properties and intra-cluster radiative transition probability of Cu_4_I_4_ NCs, thus successfully realizing single-molecule white light electroluminescence through a double-layer spin-coating device structure, as shown in Figure 9a. The brightness of the LED reaches 1500 cd m^−^^2^, which is comparable to that of the devices using mononuclear and dinuclear copper complexes. Although the PLQY of the Cu_4_I_4_ NCs is about 5%, [DBFDP]_2_Cu_4_I_4_ still endows its LED with a maximum EQE of 0.73%. In 2019, Hui Xu et al. introduced a donor group into the ligand to selectively optimize the excited state of the NCs to promote the radiative transition and fabricated the first sky-blue LED using copper iodide NCs as the emission layer, with the brightness and EQE as high as ~7000 cd m^−2^ and ~8%, respectively [86]. The electron-donating effect of the donor group in the ligand delocalizes the molecular transition orbitals from the NC unit to the ligand, effectively enhancing the ligand-centered radiative transition, and resulting in a 13-fold increase in the PLQY. In turn, the excellent rigidity and photostability of the NC units improve the color purity and efficiency stability of the device. The device structure is shown in Figure 9b. In 2022, Hui Xu et al. reported two bisphosphine-chelated asymmetric Cu_4_I_4_ NCs, [DMACDBFDP]_2_Cu_4_I_4_ and [DPACDBFDP]_2_Cu_4_I_4_, to overcome the efficiency limit of the Cu_4_I_4_ NC-based LEDs [87]. The asymmetric modification of the acridine group and the electron-donating effect led to the NCs exhibiting an excited state dominated by iodine-ligand charge transfer, resulting in a significant increase in the singlet radiation rate constant and a decrease in the triplet non-radiative rate constant. Consequently, [DPACDBFDP]_2_Cu_4_I_4_ exhibits a 16-fold (81%) increase in PLQY and a 20-fold (19.5%) increase in EQE compared to the non-functionalized parent NCs, and the device structure is shown in Figure 9c. This is already comparable to the best results for Cu^+^ complex-based OLEDs, and comparable to the top-notch efficiencies of various electroluminescent devices. This work shows that, based on rational ligand engineering, NCs’ materials can be developed into one of the most promising systems for practical electroluminescent applications in display and lighting.

The thermally activated delayed fluorescence properties (TADF) exhibited by high-emissivity copper iodide NCs have received extensive attention. Compared with mononuclear and di-nuclear Cu(I) complexes (n = 1, 2), copper iodide NCs have the advantage of being rigid and thus thermal and photostable. However, only Cu_4_I_4_ NCs are used as light-emitting layers in electroluminescent LEDs. In 2020, Xu et al. used the N^∧^P^∧^N tridentate ligand 2-[2-(dimethylamino)phenyl(phenyl)phosphorus]-N,N-dimethylaniline with a rigid structure for the first time [88]. A hexanuclear Cu(I) iodide NCs Cu_6_I_6_(ppda)_2_ with efficient single molecule white light and the highest PLQY was synthesized. The Cu_6_I_6_(ppda)_2_ NCs exhibit strong white emissions in a powder state at room temperature. Due to the relatively relaxed structure, the emission color of Cu_6_I_6_(ppda)_2_ changes from blue to white to yellow during the process it undergoes from crystal to powder to thin film. The emission of Cu_6_I_6_(ppda)_2_ originates from the contributions of metal-to-ligand charge transfer (MLCT) and halogen-to-ligand charge transfer (XLCT). The device structure of electroluminescent LEDs based on hexanuclear copper iodide NCs is shown in Figure 9d. The electroluminescent LEDs exhibit a stable yellow emission with International Commission on Illumination (CIE) coordinates of (0.43, 0.51), a maximum EQE of 0.31%, and a maximum CE of 0.85 cd A^−1^. In 2020, Marian et al. reported cationic organocopper NCs [Cu_4_(PCP)_3_]^+^ (PCP = 2,6-(PPh_2_)_2_C_6_H_3_), which have the characteristic of inhibiting nonradiative decay and a robust narrow-band green luminophore, and their PLQY is as high as 93%. The PL decay kinetics confirmed by density functional theory (DFT) calculations reveal a complex emission mechanism involving both TADF and phosphorescence contributions. Experiments found that after the NCs were prepared into electroluminescent LEDs by a solution treatment, the material had an EQE of up to 11% and a narrow emission bandwidth (65 nm FWHM). The device structure is shown in Figure 9e.

As a part of advanced visual perception, circularly polarized light has received extensive attention in three-dimensional display, biological coding, optical data processing, and other applications. In the process of using traditional physical methods to generate circularly polarized light (CPL), the unpolarized light source needs to pass through a polarizer and a quarter-wave plate in turn, resulting in a large loss of brightness. Therefore, a chiral luminescent material that can directly generate a CPL is developed, which is an effective means to solve this problem [89,98,99,100]. Nowadays, the pursuit of a CPL with both a high brightness and high asymmetry factor (glum) has become an inevitable development trend. With extensive research on chiral organic molecules and inorganic nanoparticles for amplifying CPL, copper halide hybrid NCs have attracted extensive attention due to their potential for efficient CPL.

In 2021, Yao et al. reported the first bright and efficient electroluminescent LEDs based on assembled-chirality Cu-I-phosphine hybrid NCs [90]. Chiral phosphine ligands were introduced into the copper iodide material system, and chiral R/S-Cu_2_I_2_(BINAP)_2_ binuclear copper-iodide NCs were prepared by a simple liquid-phase diffusion method. Benefiting from the bisphosphine chelation coordination of the chiral ligand, the stability of the chiral NCs was greatly improved, and it could be stably dissolved in polar solvents, such as dimethyl sulfoxide, thereby realizing the solution processing synthetic method. Using the polymer PVP K88-96 micelles as an aid, due to the strong intermolecular interaction between the chiral ligands, the liquid-phase self-assembly of chiral NCs can be achieved and results in good dispersibility in the ethanol solution. Compared with the NCs existing in a monodisperse state in polar solvents, the microcrystalline aggregates obtained by self-assembly show a significantly enhanced CPL signal, and the asymmetry factor can reach the highest order of magnitude close to 10^−2^. Moreover, the solution spin coating of R/S-Cu_2_I_2_(BINAP)_2_ chiral NCs dissolved in DMSO; the obtained self-assembled films exhibited a high degree of crystallographic orientation. Compared with the monodispersed clusters in the solvent, the aggregated films still showed obvious CPL signal enhancement and glum amplification. Electroluminescent LEDs based on R-Cu_2_I_2_ (BINAP)_2_ chiral NCs crystalline thin films exhibit a maximum brightness of up to 1200 cd m^−2^ and a maximum EQE of 0.54% under the action of an applied electric field. The device structure is shown in Figure 9f. This provides a new idea for constructing novel CPL materials and LEDs based on chiral MNCs.

## 4. Conclusions and Perspective

In the past decade, the research on electroluminescent LEDs based on MNCs has made great progress, but it is still in its infancy, and there is still much work to be done. The outlooks are as follows:Compared with traditional QDs materials, the PLQYs of MNCs is still at a low level. Co-optimization of the metal core and surface motifs needs to be employed to improve the PLQYs.The emission type of MNCs directly determines the performance of electroluminescent LEDs. At present, most of the electroluminescent LEDs are fluorescent MNCs, while phosphorescent MNCs have also been trialed and a high device efficiency has been obtained. Delayed fluorescence is an important feature of a new generation of luminescent materials. Triplet excitons can be effectively captured by the reverse intersystem crossing from the triplet to the singlet. It has a high radiation decay rate and fluorescence efficiency. In theory, the internal quantum efficiency can reach 100%, and it has been widely used in OLEDs and has made great progress. Therefore, exploring the development of delayed fluorescence MNCs’ materials and applying them as light-emitting layers to electroluminescent LEDs is bound to be of great significance for the improvement of device efficiency.Due to the imperfect structure, energy level, and emission mechanism of MNCs, the structure optimization and detailed working mechanism of MNC-based electroluminescent LEDs still need to be further investigated. Through the synergistic effect of the appropriate combination of functional layers with appropriate energy levels and the optimized light-emitting layer, the performance of the device (including the brightness and EQE) can be greatly improved. There is no doubt that with the development of high-performance LEDs, the importance of interface optimization will become more and more prominent. By adjusting the energy levels and thicknesses of the electron and hole transport layers, a relatively balanced electron and hole can be obtained in the light-emitting layer, which is beneficial to the radiative recombination of carriers in the light-emitting layer. At the same time, it is of great significance to confine the carriers in the light-emitting layer region to reduce the loss caused by non-radiative recombination. Drawing on the device design ideas, the analysis and optimization methods of OLEDs and perovskite LEDs will be conducive to developing the MNCs’ light-emitting layer LED structures and is expected to accelerate the development and device improvement of electroluminescent LEDs based on MNCs.In addition to applying the luminescence properties of MNCs to the luminescent layer of the electroluminescent LEDs, they can also be applied to the transport layer of the device to improve the carrier transmission performance or make the energy levels between the layers better matched by adjusting the energy levels of the transport layers, so as to jointly improve the performance of the device. At present, there are very few studies on improving the device transport layer based on MNCs, and researchers should further explore this field to enrich the application prospects of MNC-electroluminescent LEDs.At present, the research work based on MNCs is based on the processing and preparation of organic phase solvents, but organic solvents will cause great harm to the environment and operators. At the same time, a large number of fluorescent MNCs are water soluble, but there is no report on the application of aqueous MNCs as light-emitting layers, and there is still a large research space for MNCs in this field.Chiral metal nanoclusters have become a research hotspot in nanoscience and nanotechnology due to their potential applications in asymmetric catalysis, drug design, and chiral recognition and separation. At present, a series of optically pure monochiral NCs have been obtained through ligand engineering, and the application of these chiral MNCs in electroluminescent LED devices has promising application prospects. Some researchers have carried out related work in copper-iodine NCs and achieved excellent results, and the application of other MNCs, such as chiral Au NCs in electroluminescent LEDs, also needs further exploration.

In conclusion, there is still a long way to go for the development of large-scale commercial applications of metal nanocluster electroluminescent LEDs in the future. However, with the development of 5G networks and artificial intelligence, new fluorescent materials and next-generation LED technologies will receive increasing attention; more and more researchers are also devoted to this direction. Therefore, we firmly believe that these problems can be solved in the near future, and electroluminescent LEDs based on metal nanoclusters will also flourish in the future. The authors also hope that this paper will contribute to researchers’ understanding of relevant basic science issues and the development of relevant research areas.

## Figures and Tables

**Figure 1 nanomaterials-12-03837-f001:**
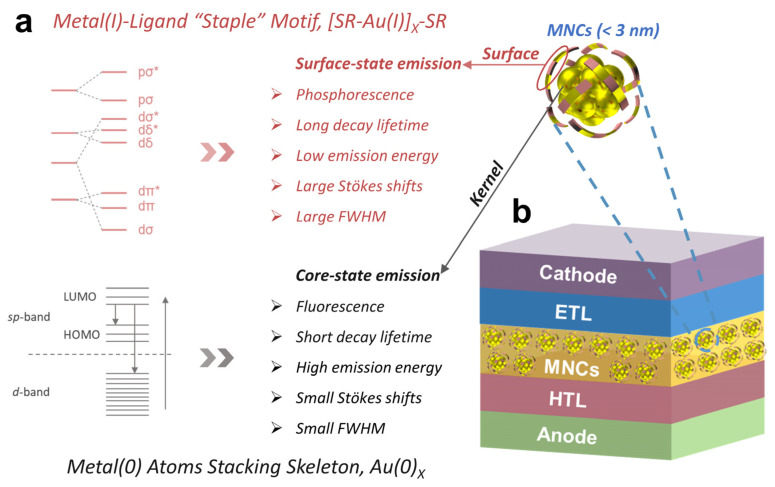
A simple model of MNCs and typical device structure of MNC-based electroluminescent LEDs. (**a**) The luminescence mechanism and characteristics of the surface metal-ligand motifs and kernel metal atoms correlated with the emission of MNCs. * = antibonding molecular orbital. (**b**) The general device structure diagram of conventional MNC-based electroluminescent LEDs.

**Figure 2 nanomaterials-12-03837-f002:**
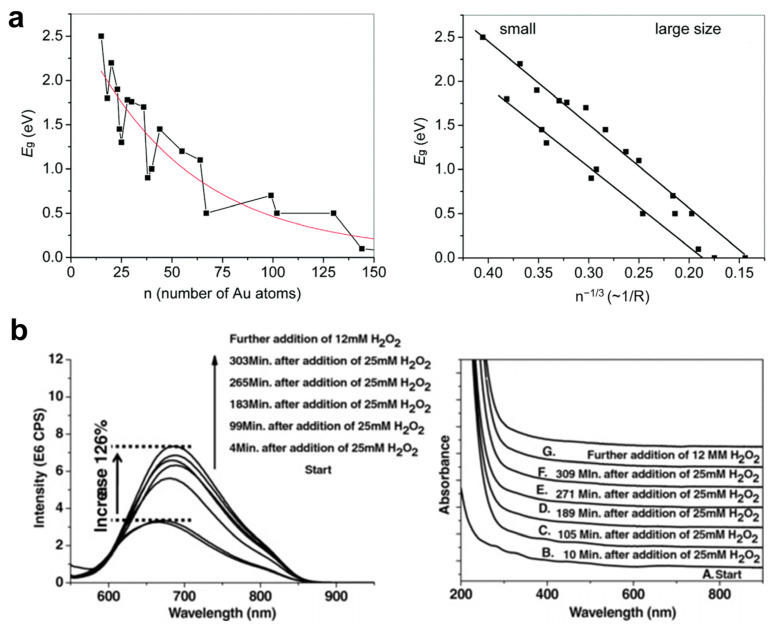
Metal core-related photoemission of MNCs. (**a**) Bandgap *E*_g_ energies as a function of the size of Au_n_(SR)_m_ NCs. Copyright (2015) Royal Society of Chemistry. (**b**) The oxidation process of Au_25_(SG)_18_ by H_2_O_2_. Copyright (2012) Wiley-VCH.

**Figure 3 nanomaterials-12-03837-f003:**
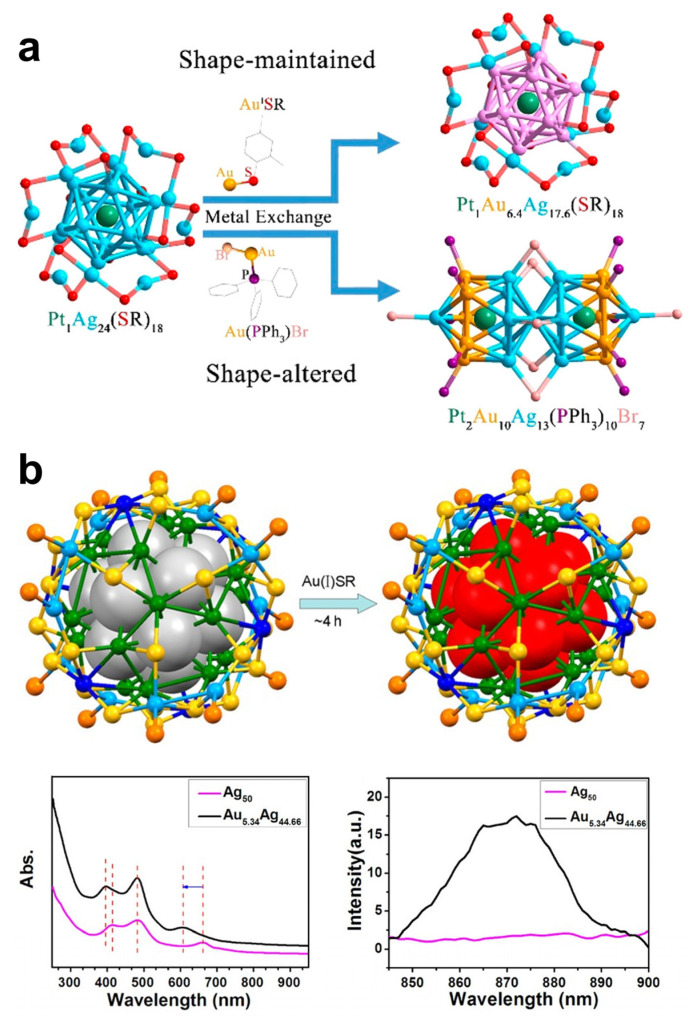
Metal doping-related photoemission of MNCs. (**a**) Shape-controlled synthesis of the sphere and rodlike PtAuAg trimetallic nanoclusters by the metal exchange method. Copyright (2016) Wiley-VCH. (**b**) The upper section is the schematic diagram of the exchange from Ag_50_(Dppm)_6_(TBBM)_30_ to Au_x_Ag_50−x_(Dppm)_6_(TBBM)_30_ by Au atom doping. The bottom section is a comparison of the UV−vis and PL spectra of Ag_50_(Dppm)_6_(SR)_30_ and Au_5_._34_Ag_44_._66_(Dppm)_6_(SR)_30_. Copyright (2017) American Chemical Society.

**Figure 4 nanomaterials-12-03837-f004:**
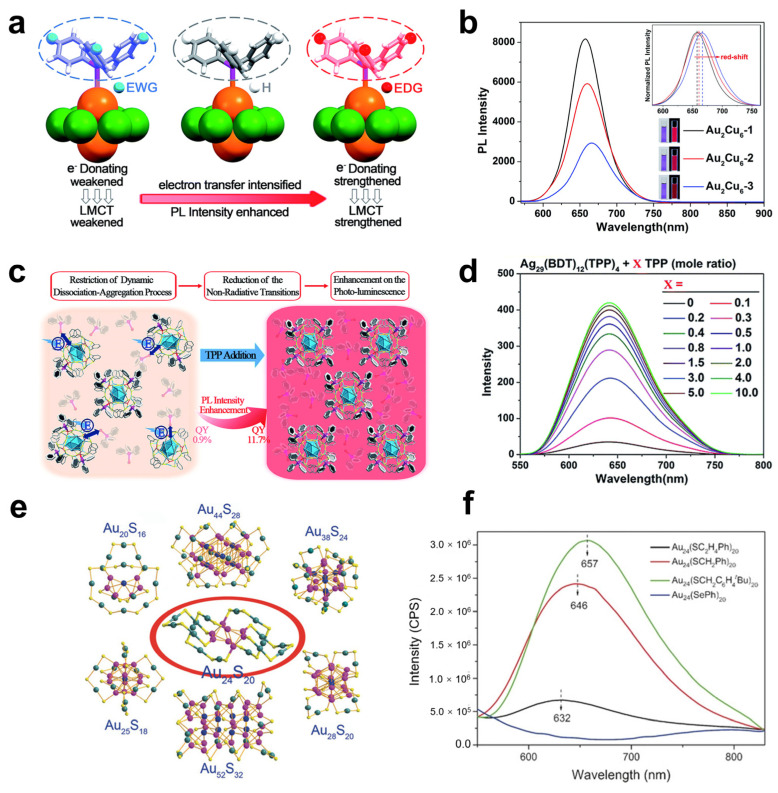
Ligand shell-related photoemission of MNCs. (**a**) The illustrations of decreased LMCT and fluorescence induced by the electron-deficient group of EWG as well as enhanced LMCT and fluorescence induced by the electron-withdrawing group of EDG. (**b**) The spectra on the PL of the two ligand-treated and pristine Au_2_Cu_6_ NCs. Copyright (2017) Royal Society of Chemistry. (**c**) Schematic illustration of the limited surface ligand dissociation process of Ag_29_(BDT)_12_(TPP)_4_ NCs after the addition of triphenylphosphine. (**d**) PL intensity variation in Ag_29_(BDT)_12_(TPP)_4_ NCs after a different amount of triphenylphosphine was added. Copyright (2018) Royal Society of Chemistry. (**e**) Structural frameworks of various Au nanoclusters. (**f**) Fluorescence spectra of four different ligand-protected Au24(SR)20 NCs. Copyright (2016) Wiley-VCH.

**Figure 5 nanomaterials-12-03837-f005:**
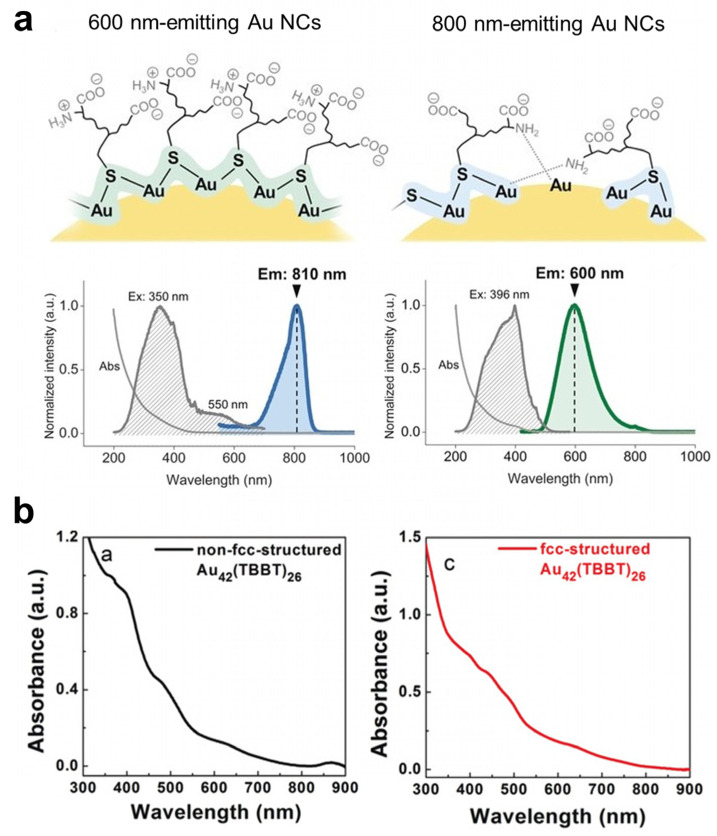
Surface motif-related photoemission of MNCs (**a**) The upper section is the schematic diagram of the surface environment of Au NCs corresponding to 600 nm and 810 nm emissions, respectively. The bottom section is the absorption, excitation, and emission spectra of Au NCs. Copyright (2016) Wiley-VCH. (**b**) Schematic structure and corresponding photoluminescence spectra of Au42(TBBT)26 with the same composition but different metal core packing modes. Copyright (2019) Wiley-VCH.

**Figure 6 nanomaterials-12-03837-f006:**
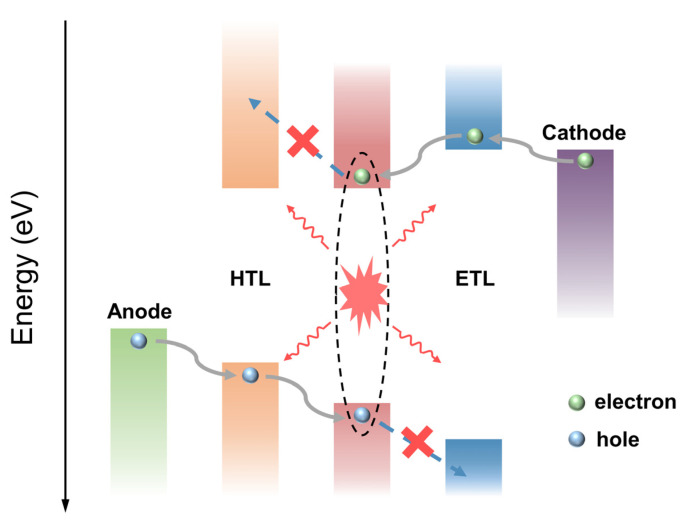
The general energy level and light-emitting mechanism diagram of MNC-based electroluminescent LEDs.

**Figure 7 nanomaterials-12-03837-f007:**
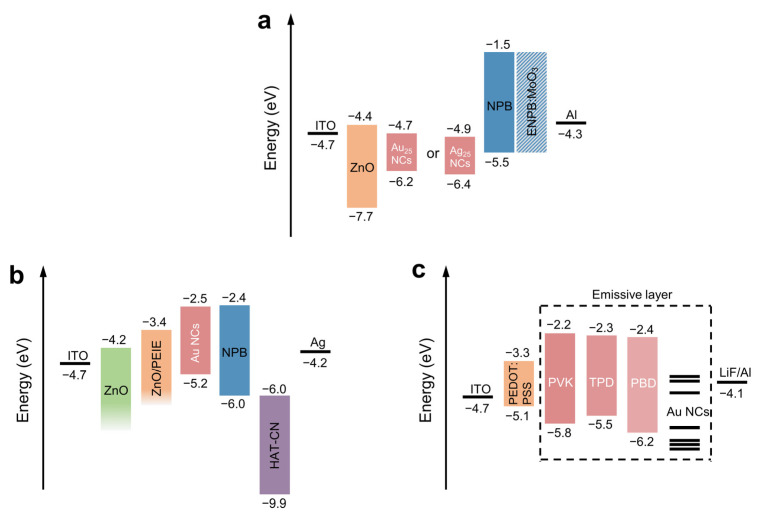
The energy level diagrams of monometallic NCs. (**a**) The energy level diagram of Au or Ag NCs-LEDs. Copyright (2014) Wiley-VCH. (**b**) The energy level diagram of Au NCs-LEDs while the Au NCs with higher PLQY. Copyright (2015) Royal Society of Chemistry. (**c**) The energy level diagram of LEDs with yellow Au NCs and a blue-emitting organic host material mixed as the light-emitting layer. Copyright (2018) Springer Nature.

**Figure 8 nanomaterials-12-03837-f008:**
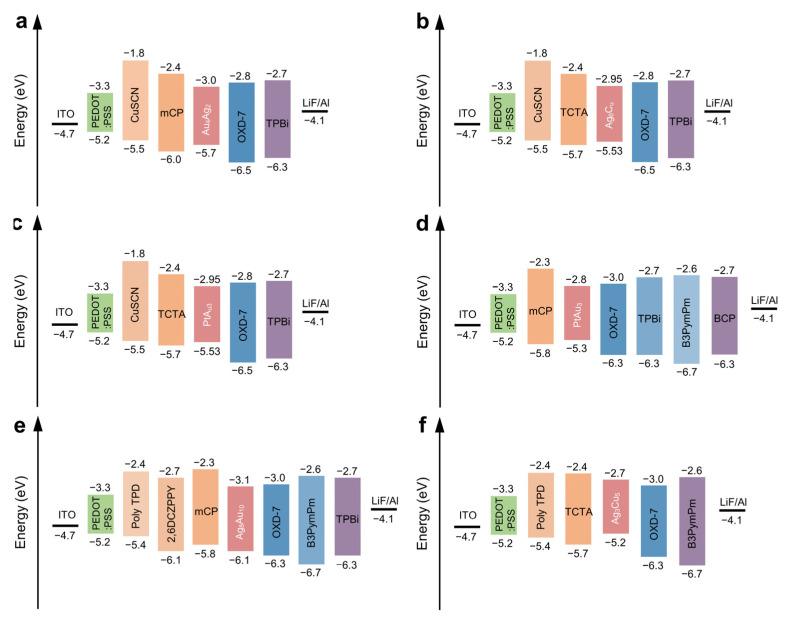
The energy level diagrams of heterometallic NCs. (**a**) The energy level diagram of Au_4_Ag_2_ NCs-LEDs. Copyright (2015) Wiley-VCH. (**b**) The energy level diagram of Ag_6_Cu NC LEDs. Copyright (2016) Royal Society of Chemistry. (**c**) The energy level diagram of PtAu_3_ NC LEDs. Copyright (2018) Royal Society of Chemistry. (**d**) The energy level diagram of tetraphosphine-PtAu_3_ NC LEDs. Copyright (2019) Royal Society of Chemistry. (**e**) The energy level diagram of Ag_8_Au_10_ NC LEDs. Copyright (2020) American Chemical Society. (**f**) The energy level diagram of Ag_3_Cu_5_ NC LEDs. Copyright (2021) American Chemical Society.

**Figure 9 nanomaterials-12-03837-f009:**
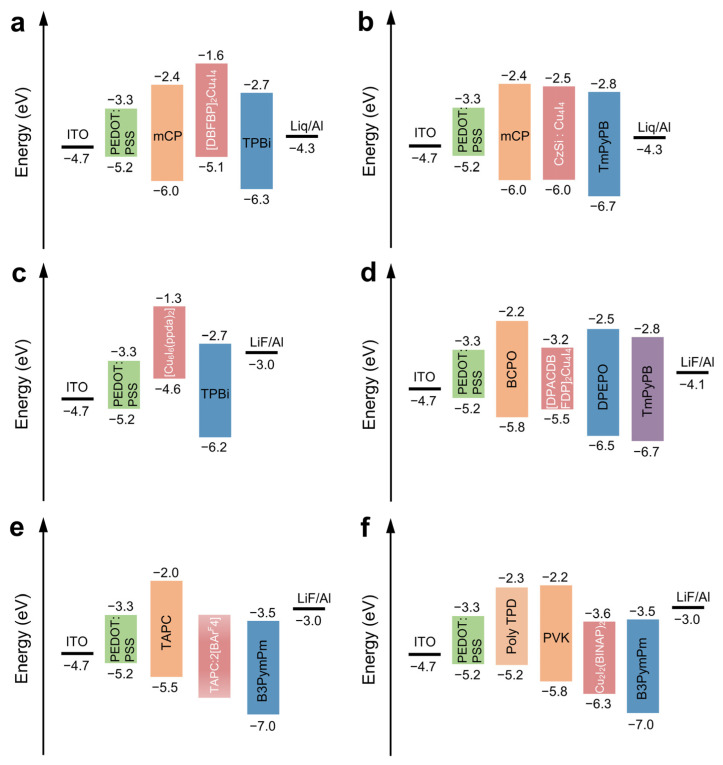
The energy level diagrams of copper iodide NCs. (**a**) The energy level diagram of [DBFBP]_2_Cu_4_I_4_ NC LEDs. Copyright (2017) American Chemical Society. (**b**) The energy level diagram of CzSi:Cu_4_I_4_ NC LEDs. Copyright (2019) American Association for the Advancement of Science (**c**) The energy level diagram of [DPACDBFDP]_2_Cu_4_I_4_ NC LEDs. Copyright (2022) American Chemical Society. (**d**) The energy level diagram of Cu_6_I_6_(ppda)_2_ NC LEDs. (**e**) The energy level diagram of TAPC:2[BArF_4_] NC LEDs. Copyright (2020) Royal Society of Chemistry. (**f**) The energy level diagram of Cu_2_I_2_(BINAP)_2_ NC LEDs. Copyright (2021) American Chemical Society.

**Table 1 nanomaterials-12-03837-t001:** The list of all abbreviations used in the text.

Definition	Abbreviations
metal nanoclusters	MNCs
nanoclusters	NCs
light emitting diodes	LEDs
quantum dots	QDs
metal nanoparticles	MNPs
highest occupied molecular orbital	HOMO
lowest unoccupied molecular orbital	LUMO
photoluminescence quantum yields	PLQYs
organic light emitting diodes	OLEDs
quantum dots light emitting diodes	QLEDs
electron transport layer	ETL
electron injection layer	EIL
hole transport layer	HTL
hole injection layer	HIL
energy gap	E_g_
ligand to metal charge transfer	LMCT
aggregation induced emission	AIE
ligand to metal-metal charge transfer	LMMCT
4-tert-butyl sulfide	TBBT
face center cubic	FCC
body centered cubic	BCC
indium tin oxide	ITO
lithium fuoride	LiF
poly(N-vinyl carbazole)	PVK
N,N′-Bis(3-methylphenyl)-N,N′-diphenylbenzidine	TPD
hexagonal close packed	HCP
cetyltrimethylammonium bromide	CTAB
external quantum efficiency	EQE
ethoxylated polyethyleneimine	PEIE
copper thiocyanate	CuSCN
N,N-dicarbazolyl-3,5-benzene	mCP
2,6-(PPh_2_)_2_C_6_H_3_	PCP
tris(diphenylphosphine)methane	CH(PPh_2_)_3_
tris(4-(9H-carbazol-9yl)phenyl)amine	TCTA
1,3-bis(5(4-(tert-butyl)phenyl)-1,3,4-oxadiazol-2-yl)benzene	OXD-7
full width at half maximum	FWHM
current efficiency	CE
3,6-di-tert-butyl-1,8-diformyl-9H-carbazole	H3decz
2,9-bis(diphenylphosphine)-dibenzofuran	DBFDP
9,9-dimethylacridine	DMAC
(dimethylamino)phenyl(phenyl)phosphino]-N,N-dimethylaniline	ppda
thermally activated delayed fluorescence	TADF
metal to ligand charge transfer	MLCT
halogen to ligand charge transfer	XLCT
International Commission on illumination	CIE
density functional theory	DFT
circularly polarized light	CPL
bis(diphenylphosphino)-binaphthalene	BINAP

**Table 2 nanomaterials-12-03837-t002:** Optical and electrical performances of the electroluminescent LEDs based on MNCs in the text. N = not reported. W = white light.

	Light−Emitting Layer	PLQY(%)	L_max_(cd m^−2^)	EQE_max_(%)	Wavelength (nm)	Ref.
Monometallic NCs	Au_25_ or Ag_25_ NCs	N	N	0.013	750	[76]
	GSH: Au NCs	15	40	0.12	625	[77]
	Au NCs	4.99	100	0.08	W	[78]
Heterometallic NCs	Au_4_Ag_2_ NCs	10.3	8804	7.0	539	[79]
	Ag_6_Cu NCs	53	184	13.9	573	[80]
	PtAu_3_ NCs	90	1000	18	588	[81]
	PtAu_3_ NCs	87.3	6539	16.6	556	[82]
	Ag_8_Au_10_ NCs	63	14,859	15.7	567	[83]
	Ag_3_Cu_5_ NCs	18	8554	14.7	585	[84]
Copper iodide NCs	[DBFDP]_2_Cu_4_I_4_ NCs	5	1500	0.73	W	[85]
	[DtBCzDBFDP]_2_Cu_4_I_4_ NCs	65	7000	7.9	491	[86]
	[DPACDBFDP]_2_Cu_4_I_4_ NCs	81	4000	19.5	500	[87]
	Cu_6_I_6_(ppda)_2_ NCs	36	N	0.31	564	[88]
	Cu_4_(PCP)_3_(BF_4_) NCs	93	8900	11.2	518	[89]
	Cu_2_I_2_(BINAP)_2_ NCs	4.7	1200	0.54	515	[90]

## Data Availability

The study did not report any data.

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
