# Peer review of "Engineering Coinage Metal Nanoclusters for Electroluminescent Light-Emitting Diodes"

_nanomaterials, 2022, doi:10.3390/nano12213837_

Round 1

Reviewer 1 Report

Dear Editor,

The review article under consideration entitled “Engineering Coinage Metal Nanoclusters for Electroluminescent Light-Emitting-Diodes” provides an summary, discussion and overview for the essential parts of photoemission in coinage metal nanoclusters and with their impacts on the performance of related electroluminescent light emitting diodes. The paper also provide a valuable demonstration of the core strategies for synthesis of coinage metal nanoclusters with high luminescent characteristics.  As an extensive review paper, associated challenges, related opportunities and foreseen outlook are also discussed in the manuscript.

After a reading of this manuscript, in Reviewers opinion, the work is self-consistent and provide deep inside to the field. The overviewing papers can be much more attractive to readers in this field. The work is properly organized. The readability of manuscript is high. Extensive list of references is a good proof to cover this interesting research area. The provided figures are almost clear, although improving their quality would be greatr for the final manuscript version.

Besides the simple reproduction of many figures, I would expect more benchmark tables and comparative discussions in the manuscript. The would be the main recommendation for this paper. Then, minor changes are suggested to the Authors.

Author Response

Response to Reviewer 1 Comments

Comments: The review article under consideration entitled “Engineering Coinage Metal Nanoclusters for Electroluminescent Light-Emitting-Diodes” provides an summary, discussion and overview for the essential parts of photoemission in coinage metal nanoclusters and with their impacts on the performance of related electroluminescent light emitting diodes. The paper also provide a valuable demonstration of the core strategies for synthesis of coinage metal nanoclusters with high luminescent characteristics. As an extensive review paper, associated challenges, related opportunities and foreseen outlook are also discussed in the manuscript. After a reading of this manuscript, in Reviewers opinion, the work is self-consistent and provide deep inside to the field. The overviewing papers can be much more attractive to readers in this field. The work is properly organized. The readability of manuscript is high. Extensive list of references is a good proof to cover this interesting research area.

Response: We are glad that the reviewer finds our work interesting and well-executed. We are also grateful for your detailed comments, which are constructive and have stimulated improvements in both the scientific contents and readability of our revised manuscript.

Comment 1: The provided figures are almost clear, although improving their quality would be greater for the final manuscript version.

Response 1: Thanks for the comment. We have improved the quality of all pictures in the text based on your advice.

Comment 2: Besides the simple reproduction of many figures, I would expect more benchmark tables and comparative discussions in the manuscript. The would be the main recommendation for this paper. Then, minor changes are suggested to the Authors.

Response 2: Thanks for the comment. We have added this part to the main text. Revision is made on Page 19, and Table 2 is provided for comparison.

Table 2. Optical and electrical performances of the electroluminescent LEDs based on MNCs in the text. N = not reported.

Overall, due to the current low PLQY, the electrical properties of the electroluminescent LEDs using monometallic nanoclusters as the light-emitting layer are poor. Meanwhile, due to the higher PLQY of heterometallic and copper iodide nanoclusters, the electrical performance of the electroluminescent LEDs has reached a good level.

Reviewer 2 Report

Engineering Coinage metal nanoclusters for electroluminescent light-emitting-diodes

 In this review work, authors have done a bibliographic review on coinage MNCs to correlate their structure and luminescence properties based on experimental findings. In addition, they collected some synthetic strategies to construct highly luminescent MNCs.

For this reviewer, the review topic is of relevance, but unfortunately this review constitutes merely a description of other’s research work without a substantial authors’ critical point of view.

Moreover, in this reviewer opinion, the manuscript is not well structured. For instance, in Section 2 which according to the title is related to MNCs structure and properties, there are two subsections about the ligand shell.

After, in Section 3, authors back to ligand shell effects with redundant information.

The beginning of Section 4 is disturbing, as well, as the title of the section is like the title of the manuscript and what is written there should rather be at the Introduction.

Finally, at the end of the manuscript, authors included a section of conclusion and perspective. However, in this reviewer opinion, this section is mainly a summary of the previous sections, together with an indefinite general conclusion.

 Additional comments are listed below.

1. Along the text, authors have been made use of several abbreviations without introducing them before which makes reading difficult: in Figure 1, one can read ETL o HTL but they are introduced several pages later. Other examples, PL in line 113, AIE several times along the text. Also, HIL or EIL, …, etc.

2. All the Figures must be improved as they are unreadable.

3. There are two figures labeled as Figure 4, making reading more difficult.

Overall, authors should improve the organization of the manuscript to make reading easier besides helping to identify gaps in knowledge on this topic to “contribute to research’s understanding”.

Thus, according to this reviewer, the manuscript does not meet the overall merit required to be published in Nanomaterials.

Author Response

Response to Reviewer 2 Comments

Comments: In this review work, authors have done a bibliographic review on coinage MNCs to correlate their structure and luminescence properties based on experimental findings. In addition, they collected some synthetic strategies to construct highly luminescent MNCs. For this reviewer, the review topic is of relevance, but unfortunately this review constitutes merely a description of other’s research work without a substantial authors’ critical point of view. Moreover, in this reviewer opinion, the manuscript is not well structured. For instance, in Section 2 which according to the title is related to MNCs structure and properties, there are two subsections about the ligand shell. After, in Section 3, authors back to ligand shell effects with redundant information. The beginning of Section 4 is disturbing, as well, as the title of the section is like the title of the manuscript and what is written there should rather be at the Introduction. Finally, at the end of the manuscript, authors included a section of conclusion and perspective. However, in this reviewer opinion, this section is mainly a summary of the previous sections, together with an indefinite general conclusion.

Response: Thanks for the comment. We are thankful for your constructive comments to better revise our manuscript.

In fact, due to the current low PLQY, the electrical properties of the electroluminescent LEDs using monometallic nanoclusters as the light-emitting layer are poor. Meanwhile, due to the higher PLQY of heterometallic and copper iodide nanoclusters, the electrical performance of the electroluminescent LEDs has reached a good level. Consequently, our view is reflected in the structure of the whole article. The main reason for the poor performance of the electroluminescent LED based on the monometallic nanoclusters at this stage is the low PLQY of the metal nanoclusters. Therefore, the method to improve the electrical performance of MNCs-based electroluminescence LEDs is to improve the PLQY of MNCs through the joint regulation of the surface ligand and the nuclear state. At present, the method to improve the PLQY of MNCs is basically to regulate the surface ligands or nuclear states alone, and few work to regulate these two parts together. In order to improve the performance of MNCs-based electroluminescence LEDs, the next step is to regulate these two parts together.

Although the metal core and ligand shell influence on the photoluminescence of metal nanoclusters has mentioned in the second part, but only briefly introduces the mechanism of the two parts on the photoluminescence, it is not detailed how these two parts are specifically adjusted to improve the emission of metal nanoclusters, in the third part, the means of regulating these two parts are detailed. In general, the second part focuses on the mechanism interpretation, the third part focuses on the specific implementation means of improving photoluminescence.

Following your suggestion, the beginning of Section 4 is placed in the Introduction. And we have added a benchmark table in the main text. Revision is made on Page 19, and Table 2 is provided for comparison.

Table 2. Optical and electrical performances of the electroluminescent LEDs based on MNCs in the text. N = not reported.

Comment 1: Along the text, authors have been made use of several abbreviations without introducing them before which makes reading difficult: in Figure 1, one can read ETL o HTL but they are introduced several pages later. Other examples, PL in line 113, AIE several times along the text. Also, HIL or EIL, …, etc.

Response 1: Thanks for the comment. We have added a table containing all the abbreviations mentioned in the main text. Revision is made on Page 4, and Table 1 is provided.

Table 1. The list of all abbreviations used in the text.

Comment 2: All the Figures must be improved as they are unreadable.

Response 2: Thanks for the comment. Following your suggestion, we have rearranged all the Figures to make them clearer and more readable.

Comment 3: There are two figures labeled as Figure 4, making reading more difficult.

Response 3: Thanks for the comment. We apologize for our carelessness and have corrected this mistake.

Reviewer 3 Report

This is an extremely nice and comprehensive paper on the possibility using metallic nano-particles for new generation LEDs.
The paper is well written but rather dense! It has high archival value and to my eyes it will constitute a reference paper for the next years. Based on excellent literature analysis, The proposed conclusions are pertinent and identify the research challenges for the . This can be seen as a roadmap for the technology.
The paper can be accepted of the present form, but there is a couple of points to take into account:
1. Figures are extremely pour qualify and thus unreadable. Please, entrance the qualify!
2. Add a glossary with all abbreviations used in the paper. At the present version, many of the acronyms are not explicated and this makes the paper difficult to read.
3. Please make a clear distinction between OLED technology and electroluminescence. At the present state of the paper the difference between those two technologies is nether blurry.

Author Response

Response to Reviewer 3 Comments

Comments: This is an extremely nice and comprehensive paper on the possibility using metallic nano-particles for new generation LEDs. The paper is well written but rather dense! It has high archival value and to my eyes it will constitute a reference paper for the next years. Based on excellent literature analysis, The proposed conclusions are pertinent and identify the research challenges for the . This can be seen as a roadmap for the technology.

Response : Thanks for your recognition of our work. We are also grateful for your constructive comments that will significantly improve our work.

Comment 1: Figures are extremely pour qualify and thus unreadable. Please, entrance the qualify!

Response 1: Thanks for the comment. We have improved the quality of all pictures based on your advice.

Comment 2: Add a glossary with all abbreviations used in the paper. At the present version, many of the acronyms are not explicated and this makes the paper difficult to read.

Response 2: Thanks for the comment. Following your suggestion, we have added this part to the main text. Revision is made on Page 4, and Table 1 is provided.

Table 1. The list of all abbreviations used in the text.

Comment 3: Please make a clear distinction between OLED technology and electroluminescence. At the present state of the paper the difference between those two technologies is nether blurry.

Response 2: Thanks for the comment. Electroluminescence is also known as field luminescence, electroluminescence phenomenon refers to a kind of luminescence phenomenon where electric energy is directly converted into light energy, which includes injection electroluminescence and intrinsic electroluminescence.(1) injection electroluminescence: the electrons and holes are injected directly by the electrodes. When the electrons and the holes are recombined, the excess energy is released in the form of light. The basic structure of injection electroluminescence is a junction diode (LED); (2) intrinsic electroluminescence: the injected electrons of the electrode accelerate inside the crystal under the external strengthening electric field, touch the luminescence center and excite or disentangle it, and the electrons return to the ground state. Among them, the organic light-emitting diode (OLED) belongs to an electric current type device, which is emitted by the carrier injection and composite phenomenon, and the principle of luminescence is the intrinsic type of electroluminescence. Overall, OLED is a branch of electroluminescence technology.

Round 2

Reviewer 2 Report

In my opinion, the manuscript is still unsuitable for publishing as it is.

 Authors have not established what they claim in the abstract, that is, “correlate the properties of MNCs with their influences on electroluminescent LED applications”. In the first pages, authors have summarized different strategies to obtained MNCs with high quantum yields which is probably necessary but not directly related to the efficiency of the LED. Only in section 4, authors have summarized some examples of MNCs within the LED applications.

Also, a LED is a semiconductor light source that emits light when current flows through it, being electroluminescence the working principle. Thus, this reviewer does not clearly understand the meaning of the classification made by authors in photoluminescence LEDs and electroluminescence LEDs.

In addition, in this reviewer opinion, along the manuscript there are redundant information. For instance, “Section 2.2. Ligand shell-related photoemission” and Section 3.1. Optimization of the ligand or “Section 3.3. Metal doping” and “Section 4.2. Heterometallic NCs”

Another weak point concerns references. There are too many dedicated to QDs which is not the topic of this review (QDs are not NCs) and to OLED rather than to MNCs, that is the topic of the review.

Furthermore, authors have written (lines 103-104) “As a new generation of lighting and display devices, LEDs have received extensive attention from scientific research and industrial circles in recent years [23-27].” This statement is not supported with appropriate references. References 23-27 were published from 1995 to 2009. Indeed, to be at the end of 2022, only two references were published this year. More precisely, only 18 of 123 references were published in the last three years.

Additional comments:

In line 91: “To realize MNCs-based electroluminescent LEDs …” this reviewer can imagine authors want to mean perform or achieve or …

Authors should review reference numbers. For instance, references 92 y 95 do not seem to correspond with the text.

Lines 736-738: “In 2017, Hui Xu et al. prepared the first electroluminescent Cu4I4 NCs (DBFDP)2Cu4I4 through ligand engineering design, which solved the problems of poor processability and weak electroactivity of Cu4I4 NCs [92]. The title of Reference 92 is “Graphene‐based nanosheets with a sandwich structure” and it is not authored by Hui Xu

Another example is:

In 2020, Xu et al. used the NPN tridentate ligand 2-[2-(dimethylamino)phenyl(phenyl)phosphorus]-N,N-dimethylaniline with a rigid structure for the first time [95].

The title of reference 95 is “Antioil Ag3PO4 nanoparticle/polydopamine/Al2O3 sandwich structure for complex wastewater treatment: dynamic catalysis under natural light” which in addition is not authored by Xu.

This reviewer cannot find reference 76, 77 etc

 Keeping all these aspects in mind, I regret my advice is to reject again the present manuscript in the present form.

Author Response

Comments: In my opinion, the manuscript is still unsuitable for publishing as it is.

Response: Thanks for the comment. We regret that you still consider this manuscript unsuitable for publication, and we have thoroughly revised the full text in response to your comments and hope that it is good enough for publication.

Comment 1: Authors have not established what they claim in the abstract, that is, “correlate the properties of MNCs with their influences on electroluminescent LED applications”. In the first pages,

authors have summarized different strategies to obtained MNCs with high quantum yields which is probably necessary but not directly related to the efficiency of the LED. Only in section 4, authors have summarized some examples of MNCs within the LED applications.

Response 1: Thanks for the comment. Many factors affect the LEDs’ current efficiency and external quantum efficiency, including the PLQY of the metal nanoclusters as the light-emitting layer, the structure of the LEDs, the balance of electron and hole injection, etc. However, at this stage, since research in this area is still relatively preliminary, the main means for researchers to improve the device performance of the researchers all focus on improving the PLQYs of the metal nanoclusters used as the light-emitting layer. As can be seen from the table below, all the external quantum efficiency of metal nanoclusters with high PLQYs as the light-emitting layer of the LEDs are relatively high. Of course, the low PLQY of metal nanoclusters as a light-emitting layer does not mean that the device performance is also low. When the energy level of the light-emitting layer and the electron and hole transport layer are well matched, the performance of the device can also be improved. However, at present, there is very little work based on this aspect. We had described the means to improve the performance of the metal nanoclusters based LEDs considering device structure optimization in the Part of the Conclusion and Perspective in the revised manuscript.

Comment 2: Also, a LED is a semiconductor light source that emits light when current flows through it, being electroluminescence the working principle. Thus, this reviewer does not clearly understand the meaning of the classification made by authors in photoluminescence LEDs and electroluminescence LEDs.

Response 2: Thanks for the comment. At present, there are two ways to prepare metal nanoclusters LEDs, one is to coat metal nanoclusters on the surface of ultraviolet or blue LED chips for assembly, using photoluminescence to excite metal nanoclusters fluorescence. The other is to make metal nanoclusters into the light-emitting layer and use electroluminescence to excite metal nanoclusters fluorescence.

The mechanism of photoluminescent metal nanoclusters LEDs is the same as that of metal nanoclusters. When the ultraviolet or blue LED chip shines, the metal nanoclusters coated on it absorb higher energy photons, and the energy of the photon is transmitted to the metal nanoclusters, the metal nanoclusters is excited, resulting in fluorescence emission.

The principle of electroluminescent metal nanoclusters LEDs is different from that of photoluminescence. When a voltage is applied to the metal nanoclusters LEDs, the electron injection layer, provides electrons to be injected into the electron transport layer, and the electrons finally reach the metal nanoclusters light-emitting layer. At the same time, the hole injection layer, provides holes to be injected into the hole transport layer, and the holes eventually reach the metal nanoclusters light-emitting layer. Finally, electrons and holes are compounded in the metal nanoclusters light-emitting layer, resulting in fluorescence emission.

Comment 3: In addition, in this reviewer opinion, along the manuscript there are redundant information. For instance, “Section 2.2. Ligand shell-related photoemission” and Section 3.1. Optimization of the ligand or “Section 3.3. Metal doping” and “Section 4.2. Heterometallic NCs”

Response 3: Thanks for the comment. We put these two parts together in response to your comment.

Comment 4: Another weak point concerns references. There are too many dedicated to QDs which is not the topic of this review (QDs are not NCs) and to OLED rather than to MNCs, that is the topic of the review.

Response 4: Thanks for the comment. We apologize for citing too many references on QDs and OLED while ignoring the proportion of this part in all references. We have reduced the number of references cited in this section.

Comment 5: Furthermore, authors have written (lines 103-104) “As a new generation of lighting and

display devices, LEDs have received extensive attention from scientific research and industrial circles in recent years [23-27].” This statement is not supported with appropriate references. References 23-27 were published from 1995 to 2009. Indeed, to be at the end of 2022, only two references were published this year. More precisely, only 18 of 123 references were published in the last three years.

Response 5: Thanks for the comment. We apologize for placing too much attention on the impact of the references and neglecting its timeliness when citing it for introducing the development background of LEDs. This issue is also prevalent throughout the full text, and we have revised the references based on your comments to ensure that most of the references have recent publication dates.

Comment 6: In line 91: “To realize MNCs-based electroluminescent LEDs …” this reviewer can imagine authors want to mean perform or achieve or …

Response 6: Thanks for the comment. We apologize for the inappropriate expression in the original text and change it to “To achieve thexcellent performance of MNCs-based electroluminescent LEDs, it is of great significance to prepare stable and highly luminescent MNCs, which arouses tremendous studies on the emission origin and underlying structure-dedicated emission mechanism in luminescent MNCs over the past decades”

Comment 7: Lines 736-738: “In 2017, Hui Xu et al. prepared the first electroluminescent Cu4I4 NCs (DBFDP)2Cu4I4 through ligand engineering design, which solved the problems of poor processability and weak electroactivity of Cu4I4 NCs [92]. The title of Reference 92 is“Graphene‐based nanosheets with a sandwich structure”and it is not authored by Hui Xu.

Response 7: Thanks for the comment. We rechecked reference 92 in the manuscript, which is listed as follow, “Xie, M.; Han, C.; Zhang, J.; Xie, G.; Xu, H. White electroluminescent phosphine-chelated copper iodide nanoclusters. Chemistry of Materials 2017, 29, 6606-6610”. This article was reported by Hui Xu et al. in 2017. The article “Graphene‐based nanosheets with a sandwich structure“ that you mentioned is cited when the structure of MNC-LEDs is described, with the reference number 69 in the article.

Comment 8: Another example is: In 2020, Xu et al. used the N∧P∧N tridentate ligand 2-[2-(dimethylamino)phenyl(phenyl)phosphorus]-N,N-dimethylaniline with a rigid structure for the first time [95].The title of reference 95 is “Antioil Ag3PO4 nanoparticle/polydopamine/Al2O3 sandwich structure for complex wastewater treatment: dynamic catalysis under natural light” which in addition is not authored by Xu.

Response 8: Thanks for the comment. I rechecked reference 95 in the manuscript, which is listed as follow, “Xu, K.; Chen, B.-L.; Zhang, R.; Liu, L.; Zhong, X.-X.; Wang, L.; Li, F.-Y.; Li, G.-H.; Alamry, K. A.; Li, F.-B. From a blue to white to yellow emitter: a hexanuclear copper iodide nanocluster. Dalton Transactions 2020, 49, 5859-5868”. This article was reported by Ke Xu et al. in 2020. The article “Antioil Ag3PO4 nanoparticle/polydopamine/Al2O3 sandwich structure for complex wastewater treatment: dynamic catalysis under natural light” that you mentioned is cited when the structure of MNC-LEDs is described, with the reference number 72 in the article.

Comment 9: This reviewer cannot find reference 76, 77 etc

Response 9: Thanks for the comment. I rechecked reference 76 and 77 in the manuscript, which is in lines 845-848 in the text. But no formatting exceptions or errors were found, and they are listed as follows:

  1. Niesen, B.; Rand, B. P. Thin Film Metal Nanocluster Light‐Emitting Devices. Advanced Materials 2014, 26, 1446-1449.
  2. Koh, T.-W.; Hiszpanski, A.; Sezen, M.; Naim, A.; Galfsky, T.; Trivedi, A.; Loo, Y.-L.; Menon, V.; Rand, B. Metal nanocluster light-emitting devices with suppressed parasitic emission and improved efficiency: exploring the impact of photophysical properties. Nanoscale 2015, 7, 9140-9146.
